# Rice Momilactones and Phenolics: Expression of Relevant Biosynthetic Genes in Response to UV and Chilling Stresses

La Hoang Anh [1], Nguyen Van Quan [1], Vu Quang Lam [2], Akiyoshi Takami [2], Tran Dang Khanh [3,4] and Tran Dang Xuan [1,5,*]

1. Transdisciplinary Science and Engineering Program, Graduate School of Advanced Science and Engineering, Hiroshima University, Hiroshima 739-8529, Japan; hoanganh6920@gmail.com (L.H.A.); nvquan@hiroshima-u.ac.jp (N.V.Q.)
2. Division of Hematology, Department of Internal Medicine, School of Medicine, Aichi Medical University, Nagakute 480-1195, Japan; quanglamvu1991@gmail.com (V.Q.L.); takami-knz@umin.ac.jp (A.T.)
3. Agricultural Genetics Institute, Pham Van Dong Street, Hanoi 122000, Vietnam; tdkhanh@vaas.vn or tdkhanh@vnua.edu.vn
4. Center for Agricultural Innovation, Vietnam National University of Agriculture, Hanoi 131000, Vietnam
5. Center for the Planetary Health and Innovation Science (PHIS), The IDEC Institute, Hiroshima University, Higashi-Hiroshima 739-8529, Japan
* Correspondence: tdxuan@hiroshima-u.ac.jp; Tel./Fax: +81-82-424-6927

**Abstract:** Momilactones A (MA) and B (MB) are known as phytoalexins which principally play a role in the rice defense system against pathogens. This is the first study revealing that MA and MB contribute to rice tolerance to environmental stresses, including ultraviolet (UV) radiation and chilling conditions. The proofs were achieved by scrutinizing the responses of rice under stresses through the expression of relevant biosynthetic genes to momilactones (MRBG) and phenolics (PRBG) and their accumulation. Accordingly, the expression tendency of PRBG was in line with that of MRBGs, which increased under UV irradiation but decreased under chilling conditions. In UV-exposed rice, the proliferation of MA and MB strongly correlated to that of salicylic and chlorogenic acids, esculetin, rutin, and fisetin. In terms of increasing chilling duration, the biosynthetic propensity of MB was consistent with that of benzoic, cinnamic, $\rho$-coumaric, salicylic, and syringic acids, quercetin, and tricin while the syntheses of MA and other compounds were reduced. The concomitant biosyntheses of momilactones with these acknowledged stress-resistant phenolics imply that momilactones might play a role as signaling molecules in the response mechanism of rice to UV and chilling stresses. Further comprehensive studies should be conducted to validate this paradigmatic finding.

**Keywords:** UV radiation; chilling stress; phenolics; momilactones; gene expression; biosynthesis; ultra-performance liquid chromatography-electrospray ionization-mass spectrometry; real-time quantitative polymerase chain reaction





## 1. Introduction

Rice is an important food crop that plays a crucial role in ensuring food security for the whole world. Rice has a beneficial consumption as it is rich in nutrients, phytocompounds, and associated biological activities. For example, phenolics found in rice have antioxidant, antidiabetic, anti-inflammatory, anti-neurodegenerative, and anti-cardiovascular disease potentials [1]. In addition, momilactones are valuable diterpenoids of rice, which display antioxidant, antitumor (leukemia and colon cancer), antidiabetic, anti-skin aging, and anti-obesity properties [2–8]. However, with the increasing threat of climate change, many adverse conditions have occurred that negatively affect rice production worldwide [9]. Therefore, protective strategies are urgently needed for rice cultivation and maintaining food security globally. For this purpose, a deep understanding of the rice response mechanism to environmental stress is integrally required to protect rice plants from the impacts of adverse conditions.

Among stress conditions, UV radiation is one of the most challenging threats to rice cultivation. The depletion of stratospheric ozone leads to increased UV radiation reaching the Earth's surface [10]. A slight augmentation of UV irradiation can cause severe consequences to rice growth, resulting in a severe reduction in rice yields [10]. Besides UV effects, rice cultivars are also extremely susceptible to low temperature since they are distributed mainly in tropical and subtropical areas. Low temperatures occurring at critical reproductive stages can severely inhibit grain quality or impede rice yield [9]. In general, UV and chilling stresses (in separation and/or in combination) can cause various adverse effects on rice plants during physiological, biochemical, and molecular processes and ultimately impair rice yield and grain quality [11]. Therefore, the research on rice tolerance against environmental stresses is a top priority in many countries worldwide. Under UV and chilling stresses, rice cultivars can suffer from reactive oxygen species (ROS), leading to oxidative stress [12]. Oxidative stress causes damage to DNA, proteins, membranes, etc., resulting in an inhibition of rice growth [10]. Therefore, the induction of ROS and antioxidant responses are an integral part of the research on rice tolerance to stresses. Regarding antioxidants, the elevated accumulation of phenolic compounds is considered to strengthen plants' antioxidant capacity, which may help improve plant performance under stresses [13]. In recent decades, a tremendous effort has been made to understand the rice response mechanisms against the adverse environment. Several studies have been conducted to clarify the correlation between phenolic compounds and antioxidant activities of different rice varieties under various biotic and abiotic stresses such as salinity, drought, extreme temperatures, and diseases [14–17]. In addition, rice diterpenoids comprising momilactones A (MA) and B (MB) were reported to contribute to rice resistance against salt, drought, weed, UV, and heavy metal stresses [18–22]. Besides, the biosynthesis of phenolics and momilactones is closely associated with the regulation of genes encoding the major biosynthetic enzymes [23,24]. However, the expressions of the related genes to the biosynthesis of phenolics (PRBG), and especially momilactones (MRBG), in rice subjected to UV and chilling stresses have been least elucidated.

In the current research, the antioxidant responses of rice under UV and chilling effects were determined. Additionally, the accumulation of phenolics and momilactones were determined using high-performance liquid chromatography (HPLC) and ultra-performance liquid chromatography-electrospray ionization-mass spectrometry (UPLC-ESI-MS). The relative expression method applying real-time quantitative polymerase chain reaction (RT-qPCR) was utilized to evaluate the expressions of PRBG and MRBGs. In essence, comprehensive information regarding the biochemical changes and their genetic causes in rice mediated by UV and chilling effects is discussed.

## 2. Materials and Methods

### 2.1. Rice Variety, Reagents, and Standard Compounds

Seeds of a Japonica rice variety, namely Koshihikari (*Oryza sativa* L.), were provided by Japan Agricultural Cooperatives (JA), Hiroshima, Japan. The solvents, including methanol and hexane, were purchased from Junsei Chemical Co., Ltd., Tokyo, Japan, and Kanto Chemical Co., Inc., Tokyo, Japan. The chemicals sodium hypochlorite (NaClO), Folin-Ciocalteu's reagent, sodium carbonate ($Na_2CO_3$), aluminum chloride ($AlCl_3$), 2,2-diphenyl-1-picrylhydrazyl (DPPH), 2,2′-azinobis-(3-ethylbenzothiazoline-6-sulfonic acid) (ABTS), potassium persulfate ($K_2S_2O_8$), sodium acetate ($CH_3COONa$), catechol, benzoic acid, caffeic acid, chlorogenic acid, cinnamic acid, $\rho$-coumaric acid, $\rho$-hydroxybenzoic acid, gallic acid, salicylic acid, syringic acid, vanillin, esculetin, rutin, fisetin, morin, quercitin, tricin, and galagin were obtained from Kanto Chemical Co., Inc., Tokyo, Japan. Pure momilactones A (MA) and B (MB) were previously isolated from rice husk in our laboratory of the Plant Physiology and Biochemistry, Graduate School of Advanced Science and Engineering, Hiroshima University, Japan. The identification and confirmation of MA and MB were presented in the previous study by Quan et al. [2].

### 2.2. UV Stress Treatment

Koshihikari rice seeds were sterilized with 0.1% NaOCl for 30 min before rinsing with distilled water several times. The seeds were germinated in distilled water at 30 °C in an incubator for 48 h. Strong germinated seeds were then placed into trays (37 cm × 25 cm × 11 cm of length × width × height) containing Yoshida solution and agar 0.5%. The seedlings were grown in a growth chamber at 28 °C with a 12 h photoperiod. After 21 days, UV irradiation (253 nm, 10 $\mu$mol/m$^2$/s; GL15, Toshiba, Tokyo, Japan) (2 h and 4 h per day for 5 days) was provided from above with a distance of 50 cm between rice plants and UV source. Control rice plants were grown in normal conditions. All treatments were performed with 3 replications. Subsequently, rice plants were collected for further experiments.

### 2.3. Chilling Stress Treatment

Koshihikari rice seedlings were prepared and placed into trays with the same protocol as mentioned above. After 21 days grown in a growth chamber at 28 °C with a 12 h photoperiod, rice seedlings were subjected to chilling stress at 6 °C (4 h and 8 h per day) for 7 days. Nontreated rice plants were used as the control. All treatments were conducted in triplicate. After that, rice plants were collected for the next experiments.

### 2.4. Sample Preparation

The collected rice seedlings were dried in an oven at 40 °C and subsequently ground into powder. Rice samples (100 g) were then extracted with 200 mL of methanol 90% for 1 week. The obtained extracts were mixed with hexane in a separator. The methanolic phase was then collected. After that, the filtered methanolic extract was concentrated by a vacuum evaporator. The dried extract was kept in a vial for the determination of antioxidant activity and the quantification of phenolic and momilactone contents.

### 2.5. Antioxidant Activity

The antioxidant ability of rice seedlings under UV and chilling stresses was determined via ABTS cation decolorization and DPPH radical scavenging assays. The detailed procedures with 3 replications were described in the research of Quan et al. [25]. The inhibition of radicals was observed as the discoloration of the final solution with samples and evaluated as the decreased percentage of absorbance compared with the negative control (MeOH) at 517 nm and 734 nm for DPPH and ABTS assays, respectively. An established dose-dependent curve (linear equation) applying different concentrations of samples was used to determine the required concentration inhibiting 50% of radicals (IC$_{50}$ values). A higher IC$_{50}$ value means weaker inhibition.

### 2.6. Total Phenolic (TPC) and Flavonoid (TFC) Contents

The TPC and TFC were determined by applying the same protocols with 3 replications presented in our previous publication [26]. The calculation of TPC was based on the recorded absorbance at 765 nm and expressed as milligrams of gallic acid equivalent per gram of sample dry weight (mg GAE/g DW). TFC was measured at 430 nm and presented as milligrams of rutin equivalent per gram of sample dry weight (mg RE/g DW).

### 2.7. Quantification of Phenolics and Flavonoids by High-Performance Liquid Chromatography (HPLC)

The quantities of phenolic compounds were determined by the HPLC system consisting of a PU-4180 RHPLC pump, LC-Net II/ADC controller, and UV-4075 UV/Vis detector (Jasco, Tokyo, Japan). A methanolic sample (5 mg/mL) with a volume of 5.0 $\mu$L was injected into the XBridge$^{\circledR}$ Shield RP18 (5 $\mu$m, 2.1 × 100 mm) column (Waters Corporation, Milford, MA, USA). Gradient elution was run with 400 $\mu$L/min of the flow rate by applying the following gradient program with 2 solvents: 95% A and 5% B over 0–2 min, followed by 30% A and 70% B over 2–12 min, subsequently changed to 0% A and 100% B over

12–22 min. From 22–34 min, the mobile phase was returned to the initial condition. In which A was 0.1% formic acid in water and B was pure acetonitrile. Phenolic compounds were determined at 280 and 350 nm. Phenolic standards (5, 10, 25, 50, and 100 µg/mL) were used to establish calibration curves for the quantification of phenolic compounds in rice based on the detected peak areas in each sample. All tests were conducted in triplicate. The outcome was presented as micrograms per gram of sample dry weight (µg/g DW).

### 2.8. Quantification of Momilactones by Ultra-Performance Liquid Chromatography-Electrospray Ionization-Mass Spectrometry (UPLC-ESI-MS)

MA and MB in rice extract were quantified based on the method described by Quan et al. [2]. In particular, the UPLC-ESI-MS system included LTQ Orbitrap XL mass spectrometers (Thermo Fisher Scientific, Waltham, MA, USA) and an electrospray ionization (ESI) source. For analysis, a methanolic sample (10 mg/mL) with a volume of 3.0 µL was injected into the ZORBAX Eclipse Plus C18 (1.8 µm, 2.1 × 50 mm) column (Agilent Technologies, Santa Clara, CA, USA). The column temperature was maintained at 25 °C. The gradient model with 2 solvents was set up as follows: solvent A was 0.1% trifluoroacetic acid in water, solvent B was 0.1% trifluoroacetic acid in acetonitrile. The gradient program was: 50% B during 0–5 min, then adjusted to 100% B during 5–10 min, subsequently maintained for 0.1 min, and another 5 min for equilibration. The flow rate was 300 µL/min. The operation was run in 15.1 min. A positive FTMS mode with a range of 100 to 800 m/z was applied for mass scanning. The presence of MA and MB in rice samples was confirmed by comparing their extracted ion chromatograms (EIC) and mass spectra of samples with those of standard momilactones. The calibration curves of MA and MB were established by applying different concentrations of momilactone standards (0.5, 1, 5, and 10 µg/mL) to determine the momilactone content in the rice seedlings. The peak areas of MA and MB detected in samples were used to calculate the amount of each compound by standard curves. All quantifications were performed in triplicate. The result was expressed as micrograms per gram of sample dry weight (µg/g DW).

### 2.9. Expression of Relevant Biosynthetic Genes to Phenolic (PRBG) and Momilactone (MRBG) by Real-Time Quantitative Polymerase Chain Reaction (RT-qPCR)

The total RNA was obtained using the RN-Sure Plant Mini Kit (Intégrale Co., Ltd., Tokushima, Japan). Single- and double-stranded DNA was removed from total RNA using DNase I, RNase-free (Thermo Scientific, Waltham, MA, USA). An amount of 1 µg of total RNA was used for synthesizing cDNA by applying a High-Capacity cDNA Reverse Transcription Kit (Applied Biosystems, Foster City, CA, USA) with an RNase Inhibitor (Applied Biosystems, Foster City, CA, USA). The obtained cDNA solution was then diluted two times for the next step. The PCR reaction was generated using KOD FX Neo (TOYOBO Co., Ltd., Osaka, Japan). The PCR program was set for initial denaturation at 96 °C for 2 min, followed by 45 cycles of denaturing at 98 °C for 15 s, annealing at 56 °C for 20 s, and extending at 72 °C for 20 s. The final extension was performed at 72 °C for 1 min.

All tests were performed in triplicate. The results of cycle threshold (Ct) values were obtained from a StepOne RT-qPCR (Applied Biosystems, Waltham, MA, USA). Based on Ct values, the relative quantifications (RQ) were calculated to compare between samples. The formula was as below:

$$RQ = 2^{-(\Delta Ct\ sample\ -\ \Delta Ct\ control)} \tag{1}$$

ΔCt sample: Ct value of tested genes in rice under stress conditions after subtracting Ct value of housekeeping gene
ΔCt control: Ct value of tested genes in rice under normal conditions after subtracting Ct value of housekeeping gene

The primers for PRBG and MRBG in the biosynthetic pathway of phenolics and momilactones [23,24] were designed by applying the NCBI Primer-BLAST program [27]. The list of primers is described in Supplementary Materials, Table S1.

### 2.10. Statistical Analysis

Minitab 16.0 software (Minitab Inc., State College, PA, USA) was used for data analyses. The results are expressed as mean $\pm$ standard deviation (SD) ($n$ = 3). One-way ANOVA using Tukey's test at $p < 0.05$ was applied to indicate significant differences between samples. Pearson's correlation coefficients among parameters were determined using the same software.

## 3. Results and Discussion

### 3.1. Total Phenolic (TPC) and Flavonoid (TFC) Contents and Antioxidant Responses of Rice Seedlings against UV and Chilling Stresses

Phenolic compounds play a crucial role in the plant defense system against biotic and abiotic stresses [28]. These substances with UV-absorbing features can be effectively determined by UV-visible spectroscopy methods [29]. Preceding studies screened a wide range of wavelengths and generated the most appropriate absorbance to quantify total phenolics (TPC) (e.g., 765 nm for the Folin-Ciocalteu method) and flavonoids (TFC) (e.g., 430 nm for the aluminum chloride colorimetric protocol) in plant extracts [25,26,29,30]. Accordingly, TPC and TFC in Koshihikari (a famous Japonica model rice variety) seedlings under UV and chilling exposures were determined following the same procedures, the results are shown in Table 1. In other reports, UV stress stimulated phenolic and flavonoid contents in rice [31]. However, in this investigation, the TPC and TFC of rice seedlings were reduced under UV irradiation. This might be due to the differences in treatment conditions among studies and genetic diversity of selected rice cultivars. In fact, different rice varieties in various growing stages exhibit dissimilar mechanisms to cope with UV exposure including tolerance and susceptibility. In particular, tolerant cultivars accumulated relatively higher TPC than susceptible ones [32]. In our study, the TPC values of Koshihikari rice seedlings in UVC, UV2, and UV4 treatments were 3.23, 2.65, and 2.92 mg GAE/g DW, respectively (Table 1). While the TFC of UV-treated rice seedlings was approximately two times lower than the control. The TFC values of UVC, UV2, and UV4 were 3.23, 1.84, and 1.73 mg RE/g DW, respectively (Table 1). For the chilling treatments, the TPC of Chi4 (2.81 mg GAE/g DW) was significantly lower than the control (4.08 mg GAE/g DW) (Table 1). The TPC of Chi8 (5.57 mg GAE/g DW) remarkably increased compared to ChiC and Chi4 (Table 1). Generally, chilling conditions enhanced rice TPC in the present study, in line with the research of Rayee et al. [16]. On the other hand, Rayee et al. [16] indicated a significant upregulation of rice TFC under chilling effects, whereas TFC reduced in our report. Notably, the TFC of Chi4 (0.13 mg RE/g DW) was 2.4 times lower than the control (0.31 mg RE/g DW) (Table 1). The TFC of Chi8 (0.28 mg RE/g DW) increased compared to Chi4 (Table 1); however, it was lower than the control.

**Table 1.** Total phenolic (TPC) and flavonoid (TFC) contents and antioxidant response of rice seedlings against UV and chilling stresses.

| Treatment | Sample | TPC (mg GAE/g DW) | TFC (mg RE/g DW) | DPPH Assay $IC_{50}$ (mg/mL) | ABTS Assay $IC_{50}$ (mg/mL) |
|---|---|---|---|---|---|
| UV | UVC | 3.23 $\pm$ 0.22 [a] | 3.23 $\pm$ 0.15 [a] | 0.81 $\pm$ 0.01 [b] | 1.05 $\pm$ 0.07 [a] |
| | UV2 | 2.65 $\pm$ 0.17 [b] | 1.84 $\pm$ 0.07 [b] | 0.91 $\pm$ 0.02 [a] | 1.14 $\pm$ 0.06 [a] |
| | UV4 | 2.92 $\pm$ 0.20 [ab] | 1.73 $\pm$ 0.04 [b] | 0.95 $\pm$ 0.02 [a] | 1.08 $\pm$ 0.15 [a] |
| Chilling | ChiC | 4.08 $\pm$ 0.16 [b] | 0.31 $\pm$ 0.01 [a] | 0.49 $\pm$ 0.02 [b] | 0.85 $\pm$ 0.02 [b] |
| | Chi4 | 2.81 $\pm$ 0.17 [c] | 0.13 $\pm$ 0.01 [c] | 0.61 $\pm$ 0.08 [a] | 1.00 $\pm$ 0.04 [a] |
| | Chi8 | 5.57 $\pm$ 0.23 [a] | 0.28 $\pm$ 0.01 [b] | 0.48 $\pm$ 0.02 [b] | 0.85 $\pm$ 0.08 [b] |

Outcome is expressed as mean $\pm$ standard deviation (SD). Samples in the same column within a treatment followed by different superscript letters indicate significant differences at $p < 0.05$. TPC, total phenolic content; TFC, total flavonoid content; GAE, gallic acid equivalent; RE, rutin equivalent; DW, dry weight; DPPH, 2,2-diphenyl-1-picrylhydrazyl; ABTS, 2,2'-azinobis-(3-ethylbenzothiazoline-6-sulfonic acid); UVC, control rice seedlings without UV treatments; UV2, UV-treated rice seedlings for 2 h per day; UV4, UV-treated rice seedlings for 4 h per day; ChiC, control rice seedlings without chilling treatments; Chi4, chill-treated rice seedlings for 4 h per day; Chi8, chill-treated rice seedlings for 8 h per day.

Regarding antioxidants, this activity can help rice plants mitigate injuries from environmental stress [12]. In the current study, the antioxidant responses of rice seedlings under UV and chilling were determined via antiradical (DPPH and ABTS) assays (Table 1). In the DPPH assay, the antiradical abilities of rice seedlings decreased when the UV duration increased. The $IC_{50}$ values of UVC, UV2, and UV4 were 0.81, 0.91, and 0.95 mg/mL, respectively (Table 1). On the other hand, rice seedlings revealed an insignificant difference in antiradical activities between the control and treatments in the ABTS assay. The $IC_{50}$ values of UVC, UV2, and UV4 were 1.05, 1.14, and 1.08 mg/mL, respectively (Table 1). In chilling treatments, the antiradical activities of Chi4 decreased ($IC_{50}$ = 0.61 and 1.00 mg/mL for the DPPH and ABTS assays, respectively), compared to the control ($IC_{50}$ = 0.49 and 0.85 mg/mL for the DPPH and ABTS assays, respectively) (Table 1). However, the antiradical ability of the rice seedlings was enhanced when the level of chilling conditions was elevated (the $IC_{50}$ values of Chi8 were 0.48 and 0.85 mg/mL for DPPH and ABTS, respectively) (Table 1). Overall, the antioxidant capacity might contribute to the response of the rice seedlings under chilling exposure. Meanwhile, UV irradiation might decrease the antioxidant capacities of rice seedlings.

*3.2. Changes in Phenolic and Momilactone Contents of Rice Seedlings under UV Treatment*

The changes in the chemical profiles of the rice seedlings in response to UV irradiation are shown in Table 2. Accordingly, a total of 17 compounds were detected in the rice samples. In which, there were 15 phenolic compounds belonging to the groups of simple phenols (catechol), phenolic acids (benzoic, caffeic, chlorogenic, $\rho$-coumaric, $\rho$-hydroxybenzoic, and salicylic acids), phenolic aldehydes (vanillin), coumarins (esculetin), and flavonoids (rutin, fisetin, morin, quercetin, tricin, and galangin). Other diterpenoids consisting of MA and MB were also found in the rice samples. The contents of identified compounds in the rice seedlings under UV irradiation and the control are shown in Table 2.

Among the detected phenolic compounds, the quantities of benzoic acid, caffeic acid, $\rho$-coumaric acid, morin, quercetin, and galangin reduced with the increasing period of UV treatment (Table 2). Significantly, the greatest inhibitions were found in the production of benzoic acid, caffeic acid, morin, and galangin. Notably, the benzoic acid and morin contents decreased by 1.8- and 1.4-fold, respectively, in UV4 compared to UVC (Table 2). Caffeic acid and galangin accounted for 65.49 and 69.14 μg/g DW, respectively, in the control. However, these compounds were not found in rice seedlings subjected to UV stress (Table 2). On the other hand, the impacts of UV irradiation might lead to the proliferation of compounds comprising catechol, chlorogenic acid, $\rho$-hydroxybenzoic acid, salicylic acid, esculetin, rutin, and fisetin. Catechol and $\rho$-hydroxybenzoic acid were found in UV2 with 7.77 and 41.22 μg/g DW, respectively; however, these compounds were not detected in UV4 and UVC (Table 2). Remarkably, salicylic acid was not found in UVC. However, this phenolic acid was notably one of the most abundant compounds in UV2 and UV4, with the contents of 852.78 and 853.39 μg/g DW, respectively (Table 2). These phenolic compounds were previously reported to contribute to rice antioxidant activity, especially salicylic acid, which is widely known to be an essential phytohormone involved in the proliferation of ROS scavenging enzymes in rice subjected to UV irradiation [10]. However, in the present research, a slight decrease in antioxidant activity was observed, so the high phenolic content might not be helping rice cultivars deal with oxidative stress (Table 1). Therefore, these compounds might contribute to the rice defense system through other metabolic pathways. For instance, accumulated phenols in plant cells play a role as protectors under the epidermal layer, thereby protecting cellular components and reducing damages to DNA and important functional enzymes under the influence of UV exposure [33]. Moreover, rice cultivars can mitigate UV damage by increasing UV-absorbing compounds, such as flavonoids and anthocyanins [10]. Accordingly, in this study, esculetin, rutin, and fisetin might be produced to counteract the negative effect of UV light, suggesting a photoprotective role of rice phenolics. The amounts of esculetin and rutin remarkably increased by 2.5- and 1.7-fold, respectively, in UV4 compared to UVC (Table 2). In addition

to phenolics, momilactones have been considered important compounds in the rice defense system to deal with UV irradiation. In particular, the quantity of MB was significantly enhanced in UV-affected rice [21]. In agreement with the previous report, the elevated contents of MA and MB were documented (from 3.6- to 6.4-fold over the control) when the UV level increased (Table 2). The findings suggest that momilactones might play a crucial role in the rice defense system against UV stress as novel phytohormones, which requires deeper confirmation.

**Table 2.** Changes in chemical contents of rice seedlings in response to UV stress.

| Compounds | Content (μg/g DW) | | |
|---|---|---|---|
| | **UVC** | **UV2** | **UV4** |
| a. Simple phenols | | | |
| Catechol | - | 7.77 ± 0.23 | - |
| b. Phenolic acids | | | |
| Benzoic acid | 2307.63 ± 54.28 [a] | 1449.23 ± 30.11 [b] | 1259.39 ± 41.05 [c] |
| Caffeic acid | 65.49 ± 6.47 | - | - |
| Chlorogenic acid | 40.27 ± 2.11 [c] | 61.32 ± 1.42 [a] | 49.25 ± 0.14 [b] |
| $\rho$-Coumaric acid | 292.69 ± 7.15 [a] | 236.61 ± 0.57 [b] | 234.56 ± 0.50 [b] |
| $\rho$-Hydroxybenzoic acid | - | 41.22 ± 5.13 | - |
| Salicylic acid | - | 852.78 ± 6.22 [a] | 853.39 ± 2.36 [a] |
| c. Phenolic aldehydes | | | |
| Vanillin | 43.08 ± 4.36 [a] | 41.56 ± 0.75 [a] | 39.90 ± 2.18 [a] |
| d. Coumarins | | | |
| Esculetin | 18.76 ± 1.19 [c] | 62.48 ± 2.93 [a] | 46.47 ± 1.66 [b] |
| e. Flavonoids | | | |
| Rutin | 228.83 ± 10.57 [c] | 499.02 ± 21.16 [a] | 382.53 ± 66.04 [b] |
| Fisetin | 87.08 ± 5.77 [b] | 301.99 ± 11.39 [a] | 92.92 ± 1.63 [b] |
| Morin | 78.44 ± 4.93 [a] | - | 54.49 ± 1.41 [b] |
| Quercetin | 155.44 ± 15.97 [a] | 144.47 ± 7.07 [a] | 141.10 ± 3.98 [a] |
| Tricin | 234.70 ± 9.54 [a] | 201.05 ± 8.16 [b] | 211.15 ± 6.21 [b] |
| Galangin | 69.14 ± 0.85 | - | - |
| f. Diterpenoids | | | |
| Momilactone A | 10.73 ± 1.13 [c] | 39.34 ± 0.55 [b] | 69.14 ± 7.44 [a] |
| Momilactone B | 5.48 ± 0.38 [c] | 19.60 ± 0.73 [b] | 26.46 ± 2.45 [a] |

The content of each compound is presented as mean ± standard deviation (SD). Different superscript letters in a row indicate significant differences at $p < 0.05$. -, not determined; DW, dry weight; UVC, control rice seedlings without UV treatments; UV2, UV-treated rice seedlings for 2 h per day; UV4, UV-treated rice seedlings for 4 h per day.

*3.3. Changes in Phenolic and Momilactone Contents of Rice Seedlings under Chilling Treatments*

The influence of chilling stress on the contents of 18 detected compounds grouped into phenolic acids (benzoic, caffeic, chlorogenic, cinnamic, $\rho$-coumaric, $\rho$-hydroxybenzoic, gallic, salicylic, syringic acids), phenolic aldehydes (vanillin), coumarins (esculetin), flavonoids (rutin, fisetin, morin, quercetin, and tricin), and diterpenoids (MA and MB) of the rice seedlings is displayed in Table 3, followed by a detailed demonstration.

**Table 3.** Changes in chemical contents of rice seedlings in response to chilling stress.

| Compounds | Content (μg/g DW) | | |
|---|---|---|---|
| | ChiC | Chi4 | Chi8 |
| a.  Phenolic acids | | | |
| Benzoic acid | 959.20 ± 22.89 [a] | 766.09 ± 3.09 [c] | 829.10 ± 11.30 [b] |
| Caffeic acid | 40.16 ± 2.19 [a] | 35.13 ± 4.46 [a] | 33.01 ± 1.32 [a] |
| Chlorogenic acid | 110.78 ± 0.03 [a] | 56.37 ± 6.04 [b] | 57.00 ± 4.22 [b] |
| Cinnamic acid | 44.53 ± 0.52 [a] | 4.91 ± 0.27 [c] | 8.99 ± 0.67 [b] |
| $\rho$-Coumaric acid | 346.72 ± 3.36 [a] | 262.69 ± 0.97 [c] | 290.83 ± 2.29 [b] |
| $\rho$-Hydroxybenzoic acid | 177.23 ± 25.85 [a] | 45.12 ± 4.24 [b] | - |
| Gallic acid | 25.64 ± 0.20 | - | - |
| Salicylic acid | 1262.93 ± 10.71 [b] | 1012.81 ± 58.17 [c] | 1402.41 ± 46.89 [a] |
| Syringic acid | 189.47 ± 3.94 [b] | 153.79 ± 3.25 [c] | 232.67 ± 12.06 [a] |
| b.  Phenolic aldehydes | | | |
| Vanillin | - | 10.35 ± 2.71 | - |
| c.  Coumarins | | | |
| Esculetin | 0.04 ± 0.00 [a] | 0.03 ± 0.00 [b] | 0.03 ± 0.00 [b] |
| d.  Flavonoids | | | |
| Rutin | 0.57 ± 0.01 [a] | 0.43 ± 0.01 [b] | 0.44 ± 0.01 [b] |
| Fisetin | 0.18 ± 0.01 [ab] | 0.19 ± 0.00 [a] | 0.15 ± 0.02 [b] |
| Morin | 0.07 ± 0.01 | - | - |
| Quercetin | 0.08 ± 0.01 [a] | 0.03 ± 0.00 [c] | 0.04 ± 0.00 [b] |
| Tricin | 0.16 ± 0.01 [a] | 0.06 ± 0.00 [c] | 0.10 ± 0.00 [b] |
| e.  Diterpenoids | | | |
| Momilactone A | 27.28 ± 1.10 [a] | 7.90 ± 0.40 [b] | 6.14 ± 0.49 [b] |
| Momilactone B | 8.75 ± 0.50 [a] | 4.04 ± 0.08 [c] | 5.68 ± 0.24 [b] |

The content of each compound is presented as mean ± standard deviation (SD). Different superscript letters in a row indicate significant differences at $p < 0.05$. -, not determined; DW, dry weight; ChiC, control rice seedlings without chilling treatments; Chi4, chill-treated rice seedlings for 4 h per day; Chi8, chill-treated rice seedlings for 8 h per day.

Based on the data presented in Table 3, the quantities of almost all detected phenolic acids in the rice plants decreased under chilling conditions in which the amounts of chlorogenic acid and cinnamic acid were dramatically reduced by 1.9 and 5.0 times, respectively, in Chi8 compared to ChiC. Similarly, the $\rho$-hydroxybenzoic acid content in Chi4 was 3.9-fold lower than ChiC. However, this compound was not detected in Chi8 (Table 3). Gallic acid was only determined in the control, whereas this compound was not detected in Chi4 or Chi8 (Table 3). In the study of Rayee et al. [16], the contents of all phenolic acids found in chilling-affected rice plants were also reduced, while the antioxidant activity was enhanced. Rayee et al. [16] hypothesized that phenolic compounds other than phenolic acids, such as phenols, polyphenols, and flavonoids, may be involved in the rice resistance to chilling conditions. However, all flavonoid quantities were reduced in our study under the impact of chilling (Table 3). Among them, the quantities of quercetin and tricin critically diminished by 2.0 and 1.6 times, respectively, in Chi8 compared to ChiC (Table 3). In addition, morin was found in the control, but not in either Chi4 or Chi8 (Table 3). The results show that flavonoids might also play a negligible role in the defense mechanisms of rice against chilling stress. On the other hand, salicylic acid had enhanced quantities in Chi8 (1402.41 μg/g DW) compared to ChiC (1262.93 μg/g DW) (Table 3). In previous reports, low temperature promoted the accumulation of endogenous salicylic acid in numerous plant species, for example, *Arabidopsis*, cereals, wheat, and grape, suggesting that this phenolic acid is involved in regulating plant responses to chilling conditions [34]. However, apart from a slight increase in the levels of endogenous salicylic acid in rice subjected to chilling stress, no direct correlation between salicylic acid treatment and the chilling

tolerance of rice has been announced. Rice plants also responded negatively to chilling conditions with the exogenous application of salicylic acid by increasing electrolyte leakage, lipid peroxidation, and decreasing the antioxidant enzyme activity [35,36]. Moreover, in this study, the content of syringic acid in Chi8 (232.67 µg/g DW) was elevated compared to the control (189.47 µg/g DW) (Table 3). However, no reports have explained how syringic acid contributes to rice resistance to chilling effects. In addition to phenolics, momilactones, as mentioned above, play an important role in the ability of rice plants to cope with different stresses [18,19,21]. This study indicated, for the first time, that MA might not contribute to the chilling tolerance of rice because of its reduced accumulation (3.5 and 4.4 times compared to the control) (Table 3). The quantity of MB in Chi4 was 2.2 times lower than ChiC. However, the amount was enhanced with the increasing duration of the chilling treatment, which was consistent with the proliferation of benzoic acid, cinnamic acid, $\rho$-coumaric acid, salicylic acid, syringic acid, quercetin, and tricin (Table 3). This result implies that MB may contribute to the physiological responses of rice to chilling stress. Therefore, forthcoming studies should be carried out to clarify this mechanism.

*3.4. Transcriptional Response Involved in Biosynthesis of Phenolics and Momilactones in Rice Seedlings under UV and Chilling Stresses*

According to the scheme presented by Anh et al. [23], the *PAL* gene encodes an important enzyme, namely phenylalanine ammonia-lyase, in the biosynthetic pathway of phenolic compounds. The major enzymes involved in the formation of momilactones consist of syn-copalyl diphosphate synthase-like; syn-pimara-7,15-diene synthase-like; 9-beta-pimara-7,15-diene oxidase-like; and momilactone A synthase-like, which are encoded by the *OsCPS4*, *OsKSL4*, *CYP99A3*, *OsMAS*, *OsMAS2* genes, respectively [24]. The expressions of targeted genes were determined by relative quantification (RQ) methods using the housekeeping gene. Particularly, selected housekeeping genes should exhibit stable expression in different tissue types, development stages, and experimental treatments. In this study, two housekeeping genes including *actin* and *eIF-4A* were examined, which are among the most stable genes commonly used for rice research with different treatment conditions [37–40]. After screening, we obtained that *actin* (Ct values ranged from 23.28 to 29.27) revealed a stronger expression than *eIF-4A* (Ct ranged from 27.40 to 31.93) in tested samples (Supplementary Materials, Table S2 and Figure S1). Ideally, an optimal housekeeping gene should have an average expression with Ct value of between 15 and 30 [41]. Hereby, we selected *actin* as the internal reference gene to evaluate the expressions (RQ values) of phenolic (PRBG)-and momilactone (MRBG)-relevant biosynthetic genes in rice seedlings treated with UV and chilling conditions (Supplementary Materials, Table S3). The changes in expression levels of the tested genes with the increasing effects of UV and chilling stresses are displayed in Figure 1.

Based on the results in Figure 1 and Supplementary Materials, Table S3, the expressions of all tested genes elevated in rice under UV effects. Remarkably, *CYP99A3* and *OsMAS* had the highest level of expression, followed by *OsMAS2*, *OsKSL4*, *OsCPS4*, and *PAL*. The RQ values of *PAL*, *OsCPS4*, *OsKSL4*, *CYP99A3*, *OsMAS*, and *OsMAS2* in UV2 were 2.047, 3.710, 5.851, 48.217, 54.932, and 33.265, respectively. While in UV4, the RQ values of these genes were 2.582, 3.852, 17.887, 58.781, 48.148, and 42.267, respectively. Of which, the expressions of *OsKSL4*, *CYP99A3*, and *OsMAS* in rice subjected to the increased level of UV irradiations were previously evaluated in an Indica rice variety Basmati 1 [39]; however, the expressions of *OsCPS4* and *OsMAS2* and the quantities of MA and MB were not presented. Additionally, the response mechanisms to UV stress may vary among different rice varieties. Therefore, this is the first study indicating the changes of MA and MB syntheses in the Japonica model rice variety Koshihikari based on their accumulation and relevant gene expressions under UV exposure.

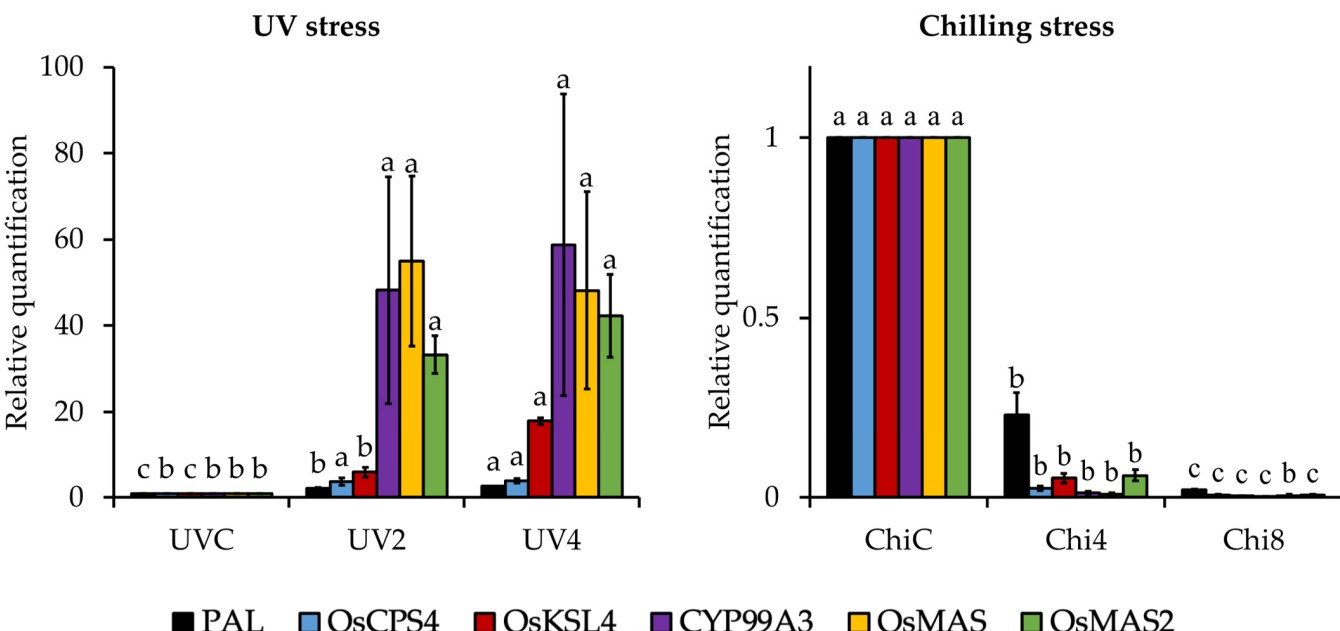

**Figure 1.** Expressions of relevant genes to the biosynthesis of phenolics and momilactones in rice seedlings against UV and chilling stresses. UVC, control rice seedlings without UV treatments; UV2, UV-treated rice seedlings for 2 h per day; UV4, UV-treated rice seedlings for 4 h per day; ChiC, control rice seedlings without chilling treatments; Chi4, chill-treated rice seedlings for 4 h per day; Chi8, chill-treated rice seedlings for 8 h per day; *PAL*, gene encoding phenylalanine ammonia-lyase; *OsCPS4*, gene encoding syn-copalyl diphosphate synthase-like; *OsKSL4*, gene encoding syn-pimara-7,15-diene synthase-like; *CYP99A3*, gene encoding 9-beta-pimara-7,15-diene oxidase-like; *OsMAS*, gene encoding momilactone A synthase-like; *OsMAS2*, gene encoding momilactone A synthase-like. Different letters (a,b,c) enclosed with columns (same colors) express significant differences at *p* < 0.05.

On the other hand, the rice responded to the increasing duration of the chilling treatment by a severe decrease in expressions of PRBG and MRBG. Particularly, the RQ values of *PAL* were 0.230 and 0.021 in Chi4 and Chi8, respectively. Furthermore, the RQ values of *OsCPS4*, *OsKSL4*, *CYP99A3*, *OsMAS*, and *OsMAS2* were 0.026, 0.054, 0.012, 0.010, and 0.061, respectively, in Chi4. Meanwhile, these values were 0.008, 0.004, 0.002, 0.006, 0.007, respectively, in Chi8 (Figure 1 and Supplementary Materials, Table S3). Notably, there has been no previous report indicating the effects of low temperature on the expressions of MRBG in correlation with their components in rice.

*3.5. Pearson's Correlation Coefficients between Antioxidant Activity, Chemical Profiles, and Relevant Gene Expressions of Rice Seedlings under UV Treatments*

The effects of UV on phenolic content and the expression of relevant genes in the phenylpropanoid pathway were previously indicated [31,42]. Of which, *PAL* is an important enzyme in the formation of cinnamic acid from phenylalanine which is the first step in a general phenylpropanoid pathway [23]. Thousands of phenolic compounds such as phenolic acids, flavonoids, anthocyanins, and lignins are derived from phenylpropanoids [43]. Therefore, enhanced activity of *PAL* is expected to increase the TPC of targeted organisms [43]. However, preceding studies showed different contributions of *PAL* to TPC which may depend on various factors such as plant species, treatment conditions, and sample preparations. For example, the upregulated *PAL* activity was consistent with the elevated TPC in cut carrots exposed to UV radiation [44]. In contrast, TPC did not show any correlation with PAL activity during the cold storage of lettuce [45], similar to the results shown in our report. This suggests that PAL plays an essential role but may not be the only factor determining the TPC in rice. Other enzymes in the biosynthetic pathway of phenolic compounds, such as cinnamic acid 4-hydroxylase (*C4H*), 4-coumaroyl:CoA-ligase

(*4CL*), chalcone synthase (*CHS*), chalcone reductase (*CHR*), etc., may also be important contributors to TPC in targeted organisms [23]. On the other hand, *PAL* may strongly correlate with specific phenolic compounds instead of TPC in plant response to different treatment conditions [46]. Accordingly, this research is the first indication that the expression of *PAL* was tightly linked to the accumulation of chlorogenic acid, salicylic acid, esculetin, and rutin in UV-exposed rice (Supplementary Materials, Table S4). Meanwhile, the proliferation of MA and MB might be controlled by the stimulated expressions of MRBG, including *OsCPS4*, *OsKSL4*, *CYP99A3*, *OsMAS*, and *OsMAS2* (Supplementary Materials, Table S4). Notably, the regulation of PRBG expression was consistent with that of MRBG expressions (Supplementary Materials, Table S4). Generally, this report confirms the contribution of chlorogenic acid, salicylic acid, esculetin, rutin, MA, and MB to the response mechanisms of rice against UV stress. However, the proliferation of these compounds had a low association with the antioxidant activities of rice treated with UV irradiation (correlation values of below 0.24), implying that they might contribute to rice tolerance against UV irradiation through other biological pathways (Supplementary Materials, Table S4). Based on a strong correlation between the biosyntheses of momilactones and the mentioned stress-resistant phenolics, MA and MB might play a role as signaling compounds in the physiological response of rice against UV irradiation.

*3.6. Pearson's Correlation Coefficients between Antioxidant Activity, Chemical Profiles, and Relevant Gene Expressions of Rice Seedlings under Chilling Treatments*

Regarding chilling stress, the antioxidant activity was closely associated with the changes in rice TPC and TFC (Table 1 and Supplementary Materials, Table S5) which may contribute to rice tolerance against low temperature during 8 h. In previous reports, the serious occurrence of oxidative stress in rice during the first 8 h of chilling stress was implicated, especially, a burst of ROS accumulation was recorded after the initial 4 h [47,48]. Whereas there was a sudden decrease in antioxidant enzymes of rice seedlings during the first 2 h of chilling treatment [47]. Under the generation of ROS during chilling exposure, rice plants may first use their existing antioxidant molecules (e.g., phenolics and flavonoids), followed by producing more of these metabolites to eliminate the damages [49]. These mentioned facts may explain the reduction in rice TPC and TFC and associated antioxidant activities under chilling conditions during the first 4 h in the current study. However, rice plants possibly started to produce more phenolics and flavonoids which may enhance the antioxidant capacities to deal with chilling stress after 8 h. On the other hand, the decreased expression of *PAL* was tightly linked to the diminished contents of almost all detected phenolics, except for syringic acid and salicylic acid (Supplementary Materials, Table S5). For momilactone biosynthesis, the reduced quantities of momilactones might be caused by the impeded expressions of MRBG (Supplementary Materials, Table S5). Furthermore, there was a linear relationship between the expressions of PRBG and MRBG (Supplementary Materials, Table S5). We suppose that chilling conditions might impede the expression of these genes in rice, resulting in the decreased quantities of almost all phenolics and momilactones. With the increasing level of the chilling duration, we found that the elevated contents of salicylic acid, syringic acid, benzoic acid, $\rho$-coumaric acid, quercitin, tricin, and MB might be involved in the strengthened antioxidant capacity (DPPH and ABTS assays) of rice seedlings against chilling stress (Supplementary Materials, Table S5). Additionally, the strong correlation between the proliferation of salicylic acid and syringic acid suggests that the synergic effect of these compounds might contribute to rice resistance to low temperatures.

*3.7. Concluding Remarks*

This is the first study to interpret the critical role of concomitant syntheses of phenolics and momilactones in rice under UV and chilling stresses based on the evidence obtained from both secondary metabolite and gene expression levels. The schematic representation of rice responses to UV and chilling influences via the accumulation of major compounds

and the expressions of relevant biosynthetic genes is summarized in Figure 2. Accordingly, the upregulated expression of *PAL* was closely associated with the enhanced contents of chlorogenic acid, salicylic acid, esculetin, and rutin under UV exposure. Meanwhile, the decreased expression of *PAL* was tightly linked to the reduced accumulation of almost all phenolic compounds, except for syringic acid and salicylic acid in rice treated with chilling conditions. Although known to be potent radical scavengers, phenolic compounds may not always represent strong antioxidant capacity of plants under adverse conditions due to the lack of effective substitution of hydroxyl (-OH) groups [50]. On the other hand, plants may utilize their endogenous phenolic compounds to combat environmental stresses through various biological pathways [10,33]. For instance, phenols accumulated under the epidermal layer can protect cellular components, DNA, and important enzymes from UV damages [33]. Flavonoids are also proliferated to absorb UV radiations, thereby helping plants mitigate injuries [33]. Some compounds, such as salicylic acid, act as phytohormones contributing to the promotion of antioxidant enzymes in the response mechanism of rice plants against unfavorable conditions [10]. In addition to phenolics, this study documented a potential role of momilactones in the rice defense system since their biosyntheses were regulated under increasing levels of UV and chilling exposures. Remarkably, MA and MB biosyntheses are enhanced under UV effects, whereas they are decreased in rice treated with 6 °C exposure. There was a linear relationship between the expressions of PRBG and MRBG in rice exposed to UV and chilling conditions. Furthermore, the proliferation of MA and MB was in line with that of chlorogenic acid, esculetin, rutin, fisetin, and salicylic acid under UV treatment. In chill-treated rice, the quantity of MB was consistent with that of benzoic acid, cinnamic acid, $\rho$-coumaric acid, salicylic acid, syringic acid, quercetin, and tricin, while the contents of MA and the remaining detected compounds were reduced. Probably, momilactones might contribute to the antioxidant ability of rice plants by neutralizing the oxidative effects of free radicals. However, these compounds revealed a moderate anti-free radical ability ($IC_{50}$ = 2.84 and 1.28 mg/mL for MA and MB, respectively against ABTS radical cations) [3]. In addition, the minor concentrations of momilactones in rice suggest that momilactones may contribute a secondary role in the total antioxidant activity of rice. Therefore, the actual role of momilactones in the antioxidant response of rice to unfavorable conditions requires further confirmation. On the other hand, the biosyntheses of momilactones were closely correlated to that of stress-resistant phenolics and flavonoids in the present research (Supplementary Materials, Tables S4 and S5). Most likely, they appear to be signaling substances inducing protection in rice against UV and chilling stresses. Thus, the potential functions of momilactones and their relationship with resistance-associated molecules in rice under stresses, including phytohormones (e.g., salicylic acid, abscisic acid, ethylene, and jasmonic acid) and functional enzymes (e.g., antioxidant enzymes and DNA damage repair enzymes), may be a promising approach that should be comprehensively scrutinized by future studies.

Our findings may contribute to future research on protecting rice cultivation. Of which, the exogenous application of interested compounds should be thoroughly evaluated to confirm their direct contributions to rice tolerance against biotic and abiotic stresses [38,51]. Preceding reports indicated the effects of salicylic acid on rice responses to UV and chilling exposures. Accordingly, rice responded positively to UV radiations with salicylic acid treatment through the enhancement of antioxidant enzymes, photosynthesis, pollen viability, leaf phenolic content, and yield [11]. In contrast, although the endogenous amount of salicylic acid was increased in rice under low temperature, the exogenous application of this substance caused negative impacts on rice tolerance against chilling conditions by increasing oxidative stress and decreasing antioxidant capacity [35,36]. On the other hand, momilactones may play an integral role in rice physiological responses to various stresses [18,19,52]. However, there has been no research on the effects of momilactone treatment due to the confined availability on the market as well as the difficulty in isolation and purification of these compounds [2]. Therefore, next investigations should focus on the contribution of exogenous application of momilactones to rice resistance against UV and

chilling stresses. Moreover, because adverse conditions simultaneously activate a combination of responsive molecules, the synergistic action of compounds, such as phenolics and momilactones, is a promising field of future research.

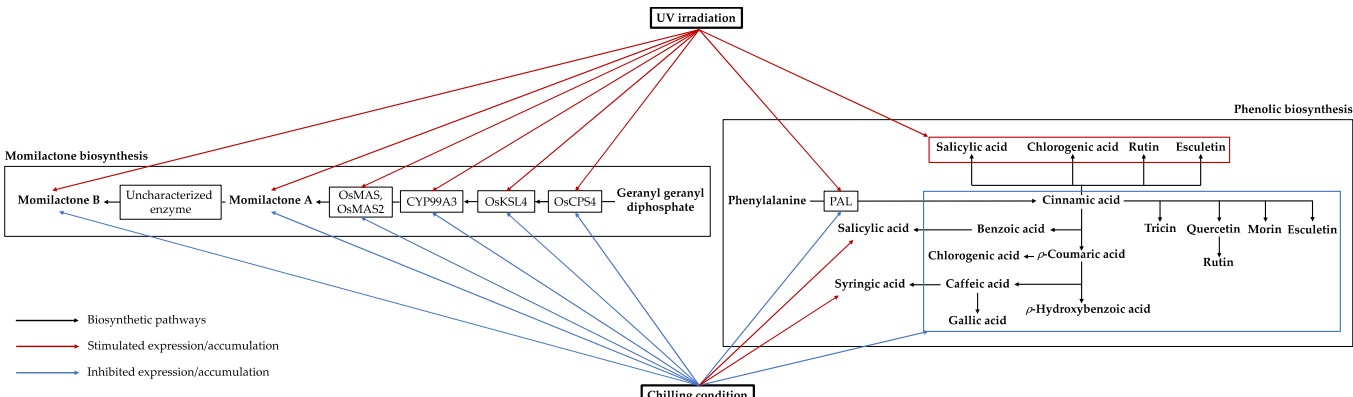

**Figure 2.** The effects of UV and chilling stresses on the accumulation of rice phenolics and momilactones and the expression of relevant biosynthetic genes. The biosynthetic pathways are aggregated following the schemes described by Anh et al. [23] and Shimura et al. [24]. *PAL*, gene encoding phenylalanine ammonia-lyase; *OsCPS4*, gene encoding syn-copalyl diphosphate synthase-like; *OsKSL4*, gene encoding syn-pimara-7,15-diene synthase-like; *CYP99A3*, gene encoding 9-beta-pimara-7,15-diene oxidase-like; *OsMAS*, gene encoding momilactone A synthase-like; *OsMAS2*, gene encoding momilactone A synthase-like.

Furthermore, genetic engineering with the support of advanced techniques and a comprehensive genome database of rice is another effective approach to improve rice tolerance against stresses focusing on the biosynthesis of secondary metabolites such as phenolics and momilactones. Many major enzymes in the biosynthetic pathway of phenolic compounds have been described [23]. Although momilactones were identified over the last several decades to exhibit valuable biological activities, the catalytic steps in the biosynthetic pathway of these compounds remain to be least elucidated. In particular, the genes encoding the hydrolyzation of MA to MB in rice are uncharacterized [53]. Clarifying these unclear aspects may lead to new research strategies for improving rice tolerance against unfavorable conditions. Moreover, expanding research on the proliferation of secondary compounds and the relevant gene expressions will not only protect rice production from the impacts of environmental stresses, but also promise other valuable achievements. For instance, phenolics and momilactones present a potential for treating various human diseases [2–5]. Therefore, the simultaneous accumulation of phenolics and momilactones in rice is promising for medicinal and pharmaceutical purposes. This study is expected to support the sustainable development goals (SDGs) of ensuring healthy lives and ending poverty and hunger globally, especially in developing countries.

**Supplementary Materials:** The following supporting information can be downloaded at: https://www.mdpi.com/article/10.3390/agronomy12081731/s1, Figure S1: A boxplot for the Ct values of candidate housekeeping genes (*eIF-4A* and *actin*) in rice under UV and chilling conditions; Table S1: Primer sequences of the tested genes; Table S2: Expression (Ct value) of candidate housekeeping genes (*eIF-4A* and *actin*) in rice under UV and chilling stresses; Table S3: Relative quantification (RQ) of relevant genes to the biosynthesis of phenolics and momilactones in rice response to UV and chilling stresses; Tables S4 and S5: Pearson's correlation coefficients between antioxidant activity, chemical profiles, and relevant gene expressions of rice seed-lings under UV and chilling stresses.

**Author Contributions:** Conceptualization, T.D.X., N.V.Q., T.D.K. and A.T.; methodology, N.V.Q., V.Q.L., and L.H.A.; investigation, L.H.A. and V.Q.L.; data curation, L.H.A. and V.Q.L.; writing—original draft preparation, L.H.A.; writing—review and editing, T.D.X., N.V.Q., A.T., V.Q.L., T.D.K. and L.H.A.; supervision, T.D.X., N.V.Q., T.D.K. and A.T.; project administration, T.D.X., N.V.Q. and A.T. All authors have read and agreed to the published version of the manuscript.

**Funding:** This research received no external funding.

**Institutional Review Board Statement:** Not applicable.

**Informed Consent Statement:** Not applicable.

**Data Availability Statement:** Not applicable.

**Conflicts of Interest:** The authors declare no conflict of interest.

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
