# Peer review of "Rice Momilactones and Phenolics: Expression of Relevant Biosynthetic Genes in Response to UV and Chilling Stresses"

_agronomy, doi:10.3390/agronomy12081731_

Round 1
Reviewer 1 Report
This article is devoted to the study of the accumulation of phenols and flavonoids, including momilactones A and B, in rice under the influence of ultraviolet stress and cold stress. In addition, the authors analyzed the expression of genes associated with the biosynthesis of phenols and flavonoids. In this paper, the authors experimentally showed the relationship between the content of phenols and momilactones and genes mediated with the biosynthesis of these substances under conditions of ultraviolet and chilling stresses. The experiments were performed at a high level, using modern research methods. The manuscript is well written and designed. However, before this manuscript can be published, some improvements should be done:
- Major Revisions
1. The authors should explain why the increase in the expression of genes mediated with the biosynthesis of phenols is associated with a significant decrease in phenols in rice under UV stress (Table 2). And why in the conducted experiments on UV stress there is a decrease in the total content of phenols, while other studies show a significant increase in the total content of phenols?
2. In my opinion, it is too brave to write that MA and MB can act as phytohormones in rice. Most likely, these substances act as antioxidants, which, due to their structure, neutralizes the oxidative effect of free radicals. The authors should delete these sentences from the article (lines 30-31, 426-427, 484-485).
- Minor Revisions
1. line 107-108: Please specify the distance between the plants and the source of UV radiation.
2. Chapter 2.9 It is necessary to remove the table and write the PCR conditions with sentences. Perhaps Table 1 should be moved to Supplementary materials. How many biological and technical repetitions of experiments were performed in the analysis of gene expression and the analysis of the content of phenols and flavonoids?
3. Chapter 3.4 and 3.5: Gene names should be italicized. For example, PAL => PAL etc.
4. In general, p < 0.05 => p < 0.05
I recommend this Ms for publication after minor revisions.
Author Response
Dear Respective Reviewer 1
Thank you very much for your valuable comments and revisions. We have revised our paper carefully, marked by red color letter and noted where and how we revised our paper. Please kindly check in the attached file.
Thank you very much
Tran Dang Xuan
Corresponding author

Reviewer 2 Report
The authors collected many data to deliberate the potential role of rice momilactone and phenolics in UV and chilling stress responses, which may correlated to how plant cells manage the ROS by antioxidant response under abiotic stress condition.
My concerns are listed as below,
1. Line 28-30: Unclear statement in Abstract, how minority compounds as momilactones to play an integral role in the physiological response of rice to UV and chilling? The quantity of momilactones or the expression of biosynthetic genes under adverse conditions may not support the importance of momilactones as direct stress responser if lacking of exogenous chemical supplement or genetic modification on biosynthetic pathway.
2. Table 2: What could the casual of Chi4 different from ChiC and Chi8 in phenolic/flavonoid content and antioxidant response. How relevant does two hours of chilling treatment to rice to trigger cold stress response?
3. Table 3, missing data in different time point about simple phenols or phenolic acids and Table 4, phenolic acid, phenolic aldehydes and flavonoids. Some of compounds were quantified in only one time but applied with statistical analysis, which data they were compared?
4. Bar chart may be better than line chart to present discontinuous data like gene expression in few time point. The table 5 and figure 1 seems to present the same results, which may be better to keep figure only in main text.
5. Table 6 may be appropriated to relocate to supplemental materials.
6. The conceptual model presenting in figure 2 showed opposite effects of two different abiotic stress on rice phenolic and momilactones expression. As both stimuli trigger ROS , what are potential scenario of phenolic and momilactones to act oppositely as anti-oxidants?
7. Did author try to exogenously supply phenolic and momilactones to rice under UV and chilling stress condition? Did chemical supplementation fit to current model reported in this manuscript?
Author Response
Dear Respective Reviewer 2
Thank you very much for your valuable comments and revisions. We have revised our paper carefully, marked by red color letter and noted where and how we revised our paper. Please kindly check in the attached file.
Thank you very much
Tran Dang Xuan
Corresponding author

Round 2
Reviewer 2 Report
Dear Authors,
I went through your cover letter and revised manuscripts, and I am appreciated your timely and significantly modified about revision. I do not have further questions and looking forward the follow-up study to address exogenous chemical supplementation to rice to directly test the potential of these compounds in future crop improvement.